# Trityl Cation-Catalyzed Hosomi-Sakurai Reaction of Allylsilane with β,γ-Unsaturated α-Ketoester to Form γ,γ-Disubstituted α-Ketoesters

**DOI:** 10.3390/molecules27154730

**Published:** 2022-07-24

**Authors:** Zubao Gan, Deyun Cui, Hongyun Zhang, Ying Feng, Liying Huang, Yingying Gui, Lu Gao, Zhenlei Song

**Affiliations:** Key Laboratory of Drug-Targeting and Drug Delivery System of the Education Ministry and Sichuan Province, Sichuan Engineering Laboratory for Plant-Sourced Drug and Sichuan Research Center for Drug Precision Industrial Technology, West China School of Pharmacy, Sichuan University, Chengdu 610041, China; ganzb@ractigen.com (Z.G.); cdy20080122115983@163.com (D.C.); zhanghy2022@163.com (H.Z.); fengyinglj@gmail.com (Y.F.); huangly823@163.com (L.H.); yingying.gui@astrazeneca.com (Y.G.)

**Keywords:** allylsilane, α-ketoester, Hosomi-Sakurai allylation, trityl cation

## Abstract

(Ph_3_C)[BPh(^F^)_4_]-catalyzed Hosomi-Sakurai allylation of allylsilanes with β,γ-unsaturated α-ketoesters has been developed to give γ,γ-disubstituted α-ketoesters in high yields with excellent chemoselectivity. Preliminary mechanistic studies suggest that trityl cation dominates the catalysis, while the silyl cation plays a minor role.

## 1. Introduction

α-Ketoesters are important synthons [1,2,3] and can be transformed into a variety of building blocks, which have found wide utility in natural products synthesis. As shown in Figure 1, a Bi(OTf)_3_-catalzyzed intermolecular cascade annulation of α-ketoesters with alkynols has been developed to construct γ-spiroketal-γ-lactones [4], a core structure in massarinoline A [5]. α-Ketoesters can also react with α-ketoacids to form isotetronic acids [6], a core structure in aspernolide A [7], by an asymmetric aldol/lactonization/enolization reaction. In this regard, development of new methods enabling an efficient synthesis of structurally diverse α-ketoesters is highly desirable.

Trityl tetrakis (pentafluorophenyl) borate [(Ph_3_C)[BPh(^F^)_4_]] [8] is well-known for providing stable and easily available carbocations. Since the pioneering work of Mukaiyama and co-workers [9], the use of trityl cations as Lewis acid catalysts has been explored in Mukaiyama aldol reactions [10,11,12,13,14,15,16], Hosomi-Sakurai allylations [17,18], Michael additions [19], Diels–Alder reactions [20,21,22,23,24,25,26,27,28,29,30,31], halogenations [32], epoxide rearrangements [27], ene reactions [33,34] and other transformations [35,36,37,38]. In these transformations, trityl cations display much higher catalytic reactivity than traditional metal-based Lewis acids. As part of our continuing interests in organosilane chemistry [39,40,41,42], we recently reported (Ph_3_C)[BPh(^F^)_4_]-catalyzed asymmetric Hosomi-Sakurai allylation of chiral crotyl geminal bis(silane) with aldehydes [43]. In this reaction, (Ph_3_C)[BPh(^F^)_4_] proved superior to traditional metal-based Lewis acids. This success led us to extend trityl cation catalysis to allylsilane-mediated reactions for which metal-based Lewis acids do not work well. We focused on β,γ-unsaturated α-ketoesters as electrophiles. Typical enones undergo Michael-type allylation [44,45,46] or [2+2] or [3+2] cyclization [47,48,49,50,51], but Ishihara and co-workers observed that Cu(NTf_2_)_2_-catalyzed reaction of β,γ-unsaturated α-ketoesters with allylsilanes gave only the inverse-electron-demand Diels-Alder (IEDDA) reaction adducts as the major products (Figure 2a)[52]. Sugimura and co-workers achieved the desired allylation using β,γ-unsaturated α,α-dimethoxy esters as the variant (Figure 2b) [53]. A stoichiometric amount of BF_3_•Et_2_O (1.1 equiv.) was required to give γ-substituted α,β-unsaturated α-methoxy esters as a mixture of *Z*/*E* isomers.

Here we report a (Ph_3_C)[BPh(^F^)_4_]-catalyzed Hosomi-Sakurai allylation of allylsilanes with β,γ-unsaturated α-ketoesters (Figure 2c). The trityl cation shows high catalytic efficiency, giving the γ,γ-disubstituted α-ketoesters in high yields with excellent chemoselectivity. Mechanistic studies suggest that silyl cation catalysis is not a major pathway. Instead, the reaction most likely proceeds via trityl cation catalysis, although we cannot completely rule out Brønsted acid catalysis.

## 2. Results and Discussion

### 2.1. Synthesis

We initially screened the metal-based Lewis acid catalysts using β,γ-unsaturated α-ketoester **1a** and allytrimethylsilane **2a** as model substrates in CH_2_Cl_2_ at 25 °C (Table 1). In the presence of 10 mol % of TiCl_4_, inverse-electron-demand Diels-Alder adduct **(±)-4a** was obtained in 97% yield; no desired Hosomi-Sakurai allylation product **3a** was detected (entry 1). SnCl_4_, AlCl_3_ and FeCl_3_ provided a mixture of **3a** and **(±)-4a**, in which **3a** was the minor isomer and the **3a**:**(±)-4a** ratio ranged from 23:77 to 45:55 (entries 2–4). BF_3_•Et_2_O and TMSOTf proved to be ineffective catalysts, leading to less than 30% conversion even after 4 days (entries 5 and 6). We also tested lanthanide-based Lewis acids such as Sc(OTf)_3_ and Yb(OTf)_3_ (entries 7 and 8). Sc(OTf)_3_ afforded undesired **(±)-4a** as the sole detectable product; no reaction occurred using Yb(OTf)_3_. In sharp contrast to metal-based Lewis acids, the trityl salt (Ph_3_C)[BPh(^F^)_4_] displayed excellent catalytic ability, generating **3a** as a single chemoisomer in 97% yield (entry 9). In fact, 1 mol % of (Ph_3_C)[BPh(^F^)_4_] was efficient enough to provide **3a** in comparably high yield and selectivity (entry 10).

With the optimal reaction conditions in hand, we examined the scope of β,γ-unsaturated α-ketoesters (Figure 3). Reactions of aryl- substituted ketoesters gave rise to **3b**–**3j** in excellent yields. The reaction tolerated substrates containing functionalized phenyl rings, naphthyl rings or heterocycles. An electron-donating substitution on the phenyl ring slightly decreased chemoselectivity, as shown in **3e** (H-S/D-A = 90:10) and **3f** (H-S/D-A = 95:5). The reaction generated **3k** and **3l** from the corresponding alkyl-substituted ketoesters. Allylation of dienyl ketoester with **2a** gave **3m** in 80% yield, with 1,4-regioselectivity dominating over 1,6-regioselectivity. β,γ-unsaturated α-ketimine ester performed well in the reaction, giving enamido ester **3n** in 75% yield. Interestingly, propargyl α-keto ester underwent 1,2-allylation exclusively, leading to α-tertiary hydroxy ester **3o** in 93% yield. Switching the ester group from OMe to a bulkier *i*-Pr group decreased chemoselectivity (**3p**, H-S/D-A = 91:9).

Trimethylallylsilanes **2b**–**2e** bearing alkyl or aryl substituents at the 2-position reacted well with **1a**, giving **3q**–**3t** in 85–96% yields (Table 2, entries 1–4). The high catalytic ability of (Ph_3_C)[BPh(^F^)_4_] also allowed facile *anti*-SE’ allylation of the bulky 3,3-dimethyl-1-trimethylallylsilanes **2f** with **1a** (entry 5). However, this catalyst did not efficiently control the diastereoselectivity of allylation: the reaction of *Z*-crotyltrimethylsilane **2g-*Z*** afforded **3v** in 96% yield but as a 3:2[54] mixture of *anti*- and *syn*-diastereomers (entry 6). A similar ratio of 3:1 was obtained using *E*-crotyltrimethylsilane **2g-*E*** (entry 7).

The trimethylsilyl enol ethers that initially formed in the reactions shown in Figure 3 and Table 2 were difficult to isolate because of their instability. Switching the silyl moiety from Me_3_Si to a bulkier Et_3_Si group in allylsilane **2h**–**2j** led to formation of the stable silyl enol ethers **5a**–**5c** in good to high yields (Figure 4). The *Z*-silyl enol ether was favored either as a single isomer (**5a** and **5c**) or the major isomer (**5b**).

### 2.2. Mechanistic Investigations

Some mechanistic investigations have been performed for trityl cation catalysis by different groups, but the results appear to be contradictory, particularly in the case of reactions involving allylsilanes or silyl enol ethers. For example, three catalytic species have been suggested for Mukaiyama aldol reactions. Denmark [14] and Mukaiyama [9,10,11,12,13,17,19] proposed the catalytic species to be a trityl cation. In this path, intramolecular transfer of the silyl group releases the product and regenerates the trityl cation catalyst. Bosnich [18] and Chen [16] proposed the catalytic species to be a silyl cation, which is a stronger Lewis acid than trityl cation. In another case, Kagan [20,21,22] proposed the catalytic species to be a Brønsted acid, potentially generated by decomposition of the trityl cation.

The accessibility of silyl enol ethers allowed us to perform detailed mechanistic investigations for our reaction (Figure 5). Allylsilanes **2a**, **2i**, **2h** and **2b** were reacted separately with β,γ-unsaturated α-ketoester **1a**. In the merged ^1^H NMR spectra of the resulting crude silyl enol ethers **5d**, **5b**, **5a** and **5e**, we were able to clearly distinguish the H^a^ signals of the different products (Figure 5(b1)). Therefore, we reacted a mixture of **2a** (1.2 equiv.) and **2i** (1.2 equiv.) with **1a** (2.0 equiv.) in one pot (Figure 5a). A mixture of **5d**, **5b**, **5a** and **5e** was generated in a ratio of 93(**5d** + **5b**):7(**5a** + **5e**) (Figure 5(b2)). We attribute the formation of **5a** and **5e** to crossed silyl cation catalysis. This result implies that 7% of **5d** and **5b** may form via silyl cation catalysis, meaning that the ratio of trityl to silyl cation catalysis should be approximately (93−7):(7+7) or 86:14. Next we reacted a mixture of **2h** (1.2 equiv.) and **2b** (1.2 equiv.) with **1a** (2.0 equiv.) in one pot. A mixture of **5d**, **5b**, **5a** and **5e** was generated in a ratio of 6(5d + 5b):94(5a + 5e) (Figure 5(b3)). The ratio of trityl to silyl cation catalysis in this reaction should be (94−6):(6+6) or 88:12. The results from these two control reactions suggest that silyl cation catalysis occurs but makes a minor contribution to our results.

Brønsted acid catalysis is another competing catalytic pathway, which we cannot rule out currently. This pathway seems unlikely to make a major contribution based on our observations (Figure 5c) that in the presence of 1.0 equiv. of Ph_3_C^+^•BF_4_^−^, the desired allylation product **3a** was obtained in 45% yield, while the by-product **(±)-4a** also formed in 45% yield. However, neither **3a** nor **(±)-4a** was detected when 1.0 equiv. of HBF_4_ was used.

Based on these results, we propose a trityl cation-based catalytic mechanism (Figure 5d). Activation of the ketone in β,γ-unsaturated α-ketoester **1** by trityl cation generates **6** [55]. Subsequent allylation with allylsilane may occur via an unusual closed transition state **7**, which allows internal C-to-O silyl transfer to give the non-crossed silyl enol ether **5** as the major product (Figure 5(b2,b3)). This also regenerates the trityl cation and catalyzes the next cycle.

## 3. Materials and Methods

### 3.1. General Procedures for the Synthesis of γ,γ-Disubstituted α-Ketoesters **3a**–**3v**

To a solution of β,γ-unsaturated α-ketoester **1** (0.11 mmol) and allylsilane **2** (0.13 mmol) and in anhyd. CH_2_Cl_2_ (2 mL) under argon atmosphere was added (Ph_3_C)[BPh(^F^)_4_] (1.0 mol %) at 25 °C. After stirring for 10 min, the reaction was quenched with *p*-TsOH (0.5 M in MeOH, 0.1 mL). The mixture was directly concentrated under reduced pressure. Purification of the crude residue via silica gel flash column chromatography (gradient eluent: 0–2.0% of EtOAc/petroleum ether) afforded γ,γ-disubstituted α-ketoester **3**. The characterization data for all synthetic compounds are provided in the Appendix A.

Methyl 2-oxo-4-phenylhept-6-enoate (**3a**): ^1^H NMR (400 MHz, CDCl_3_) δ 2.37 (dd, 1H, *J*_1_ = 7.2 Hz, *J*_2_ = 14.0 Hz), 2.45 (dd, 1H, *J*_1_ = 7.2 Hz, *J*_2_ = 14.0 Hz), 3.19 (d, 2H, *J* = 7.2 Hz), 3.34 (dddd, 1H, *J* = 7.2 Hz), 3.80 (s, 3H), 5.00 (d, 1H, *J* = 8.0 Hz), 5.03 (d, 1H, *J* = 15.2 Hz), 5.66 (m, 1H), 7.20 (m, 3H), 7.28 (d, 1H, *J* = 3.6 Hz), 7.31 (d, 1H, *J* = 7.6 Hz); ^13^C NMR (100 MHz, CDCl3) δ 40.2, 40.8, 44.9, 52.9, 117.3, 126.7, 127.5, 128.5, 135.8, 143.2, 161.3, 192.9; IR (neat) cm^−1^ 3029, 2954, 2923, 1731, 1442, 1277, 1253, 1089, 919; HRMS (MALDI, *m/z*) calcd for C_14_H_16_O_3_Na (M+Na)^+^: 255.0992, found 255.0989.

### 3.2. General Procedures for the Synthesis of Silyl Enol Ethers **5a**–**5e**

To a solution of β,γ-unsaturated α-ketoester **1** (0.22 mmol) and allylsilane **2** (0.26 mmol) and in anhyd. CH_2_Cl_2_ (4 mL) under argon atmosphere was added (Ph_3_C)[BPh(^F^)_4_] (1.0 mol %) at 25 °C. After stirring for 10 min, the reaction was quenched with NEt_3_ (1.32 mmol). The mixture was directly concentrated under reduced pressure. Purification of the crude residue via silica gel flash column chromatography (gradient eluent: 0–2.0% of EtOAc/petroleum ether) afforded silyl enol ether **5**. The characterization data for all synthetic compounds are provided in the Appendix A.

Methyl (*Z*)-4-phenyl-2-((triethylsilyl)oxy)hepta-2,6-dienoate (**5a**): ^1^H NMR (600 MHz, CDCl_3_) δ 0.79 (q, 6H, *J* = 7.8 Hz), 1.05 (t, 9H, *J* = 7.8 Hz), 2.56 (m, 2H), 3.82 (s, 3H), 3.99 (m, 1H), 5.05 (d, 1H, *J* = 10.2 Hz), 5.11 (d, 1H, *J* = 16.8 Hz), 5.78 (m, 1H), 7.30 (m, 3H), 7.38 (m, 2H); ^13^C NMR (100 MHz, CDCl_3_) δ 5.5, 6.8, 40.6, 41.8, 51.9, 116.4, 124.6, 126.4, 127.5, 128.5, 135.0, 140.1, 143.3, 165.3; IR (neat) cm^−1^ 2955, 2878, 1727, 1642, 1439, 1372, 1266, 1232, 1143, 1010, 913; HRMS (MALDI, *m/z*) calcd for C_20_H_30_O_3_SiNa (M+Na)^+^: 369.1856, found 369.1856.

## 4. Conclusions

In summary, we have developed a (Ph_3_C)[BPh(^F^)_4_]-catalyzed Hosomi-Sakurai allylation of allylsilanes with β,γ-unsaturated α-ketoesters. Various γ,γ-disubstituted α-ketoesters α-ketoesters were synthesized in high yields with excellent chemoselectivity. Mechanistic studies suggest that the trityl cation dominates catalysis, while the silyl cation plays only a minor role. Verification of this mechanism also makes the trityl cation-catalyzed asymmetric reaction possible, which is a challenging task and little progress has been achieved. The related work is ongoing in our group.

## Data Availability

The data are available on request from the corresponding authors.

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
