# Peer review of "Trityl Cation-Catalyzed Hosomi-Sakurai Reaction of Allylsilane with β,γ-Unsaturated α-Ketoester to Form γ,γ-Disubstituted α-Ketoesters"

_molecules, 2022, doi:10.3390/molecules27154730_

Round 1

Reviewer 1 Report

This work describes interesting results in the field of covalent organocatalysis. Scientific presentation looks very good, but I recommend some additional experiments to improve the mechanistic study. Since the general idea of this work is the catalytic effect of Ph3C+ species, it would be better to make 1H and/or 13C{1H} NMR titrations of Ph3C+ by the model reaction substrates. On the one hand, it allows the authors to estimate the binding constants or the reaction substrates with the catalyst. On the other hand, analysis of the NMR shifts during titration might make it possible to conclude the binding mode of Ph3C+ to the reaction substrates: it can bind to the O atom by the central C atom or para-C atom of one of the phenyl ring (coupling with the gem-, o-, or m-C seems to be less possible).

Minor comments:

1) It is better to avoid use dot between the cation and anion. I recommend to use the following writing (Ph3C)[BPh(F)4], where (F) – F in upper case.

2) Fourth line above Scheme 2: should be "disubstituted'.

3) Table 1, Scheme: two enantiomers of 4a should be represented, because a racemic mixture should be obtained.

Reviewer 2 Report

The authors reported catalytic Hosomi-Sakurai reaction of allylsilane with unsaturated alfa-ketoesters mediated by trityl borate. This catalytic system provided high chemoselectivity between allylation and hetero Diels-Alder reactions. In addition, the authors also isolated silyl enol ethers and performed control experiments, which gave some insight into a reaction mechanism. The results described here will be of interest to the readers of Molecules. I recommend this contribution for publication.

Minor points

(1) In Table 3, “R2” in structural formula of 1 and 2 should be removed because of no example of a beta-substituted compound.

(2) In Table 4, compounds 2e and 3t had 2-naphtyl group, but in Supporting information, 1-naphtyl group was described in structural formula of those compounds.

(3) In Supporting information, “General Methods” mentioned X-ray analysis, UV, and Optical rotation, which did not provide any date.  

Round 2

Reviewer 1 Report

In the fact, the authors did not make sufficient titration experiments (the represented spectra of 1:10 mixtures are not the titration experiments, unfortunately), but this was just a suggestion for improvement of this work, which is good enough even without these experiments. So, I recommend to accept this work, but still recommend the authors to consider NMR titrations in the future. In the field of organocatalysis, for the sufficient titration experiments with determination of binding constants see, for example, recent works of Paul Beer et al., Stephan Huber et al. or Dmitrii Bolotin et al.